# Evolution of Terpene Synthases in Orchidaceae

**DOI:** 10.3390/ijms22136947

**Published:** 2021-06-28

**Authors:** Li-Min Huang, Hsin Huang, Yu-Chen Chuang, Wen-Huei Chen, Chun-Neng Wang, Hong-Hwa Chen

**Affiliations:** 1Department of Life Sciences, National Cheng Kung University, Tainan 701, Taiwan; limin925@gmail.com (L.-M.H.); n34545@gmail.com (H.H.); faseno@gmail.com (Y.-C.C.); wenhueic005@gmail.com (W.-H.C.); 2Orchid Research and Development Center, National Cheng Kung University, Tainan 701, Taiwan; 3Department of Life Sciences, Institute of Ecology and Evolutionary Biology, National Taiwan University, Taipei 106, Taiwan; LEAFY@ntu.edu.tw

**Keywords:** terpene synthase, Orchidaceae, evolution, phylogenetic tree

## Abstract

Terpenoids are the largest class of plant secondary metabolites and are one of the major emitted volatile compounds released to the atmosphere. They have functions of attracting pollinators or defense function, insecticidal properties, and are even used as pharmaceutical agents. Because of the importance of terpenoids, an increasing number of plants are required to investigate the function and evolution of terpene synthases (*TPS*s) that are the key enzymes in terpenoids biosynthesis. Orchidacea, containing more than 800 genera and 28,000 species, is one of the largest and most diverse families of flowering plants, and is widely distributed. Here, the diversification of the *TPS*s evolution in Orchidaceae is revealed. A characterization and phylogeny of *TPS*s from four different species with whole genome sequences is available. Phylogenetic analysis of orchid *TPS*s indicates these genes are divided into *TPS-a*, *-b*, *-e/f,* and *g* subfamilies, and their duplicated copies are increased in derived orchid species compared to that in the early divergence orchid, *A. shenzhenica*. The large increase of both *TPS-a* and *TPS-b* copies can probably be attributed to the pro-duction of different volatile compounds for attracting pollinators or generating chemical defenses in derived orchid lineages; while the duplications of *TPS-g* and *TPS-e/f* copies occurred in a species-dependent manner.

## 1. Introduction

Terpenoids are the largest group of natural metabolites in the plant kingdom, including more than 40,000 different compounds, and have multiple physiological and ecological roles. Terpene metabolites are not only essential for plant growth and development (e.g., gibberellin phytohormones), but also important intermediaries in the various interactions of plants with the environment [1]. For example, chlorophylls and carotenoids are photosynthetic pigments, while brassinosteroids, gibberellic acid, and abscisic acid are plant hormones [2,3]. Terpenoids can be classified based on the number of isoprene units, such as hemiterpene (C5), monoterpene (C10), sesquiterpene (C15), diterpene (C20), sesterterpene (25), triterpene (C30), sesquarterpene (C35), and tetraterpene (C40) (Gershenzon and Dudareva, 2007). The increased number of cyclizations, possibly from a precursor with five additional carbon atoms, gives structural diversity. Terpenoid structures are extremely variable and most of them are low molecular weight like monoterpene (C_10_), sesquiterpene (C_15_), and diterpene (C_20_) [4]. The approximate number of monoterpenes is 1000 and more than 7000 sesquiterpenes [5].

Terepene synthases (TPSs) are key enzymes in terpenoids biosynthesis. To date, TPSs have been studied in several typical plant genomes, such as *Arabidopsis thaliana* (Arabidopsis, 32 *TPS*s) [6], *Physcomitrella patens* (earthmoss, 1 *TPS*) [7], *Sorghum bicolor* (Sorghum, 24 TPSs) [8], *Vitis vinifera* (grape, 69 TPSs) [9], *Solanum lycopersicum* (tomato, 29 TPSs) [10], *Selaginella moellendorffii* (spikemoss, 14 TPSs) [11], *Glycine max* (soybean, 23 TPSs) [12] *Populus trichocarpa* (poplar tree, 38 TPSs) [13], *Oryza sativa* (rice, 32 TPSs) [14], and *Dendrobium officinale* (Dendrobium orchid, 34 TPSs) [15]. According to the classification principle, TPSs can be generally classified into seven clades or subfamilies: *TPS-a*, *TPS-b*, *TPS-c*, *TPS-d*, *TPS-e/f*, *TPS-g*, and *TPS-h* [16]. *TPS-a*, *TPS-b*, and *TPS-g* are angiosperm-specific subfamilies, while the *TPS-e/f* subfamily is present in angiosperms and gymnosperms. *TPS-c* exists in land plants. *TPS-d* is a gymnosperm-specific subfamily, and the *TPS-h* subfamily only appears in *Selaginella moellendorffii* [16].

The full length of plant *TPS*s has three conserved motifs on C- and N-terminal regions. The conserved motif of N-terminal domain is R(R)X_8_W (R, arginine, W, tryptophan and X, alternative amino acid) and the C-terminal domain contains two highly conserved aspartate-rich motifs. One of them is the DDXXD motif, which is involved in the coordination of divalent ion(s), water molecules, and the stabilization of the active site [17,18,19]. The second motif in the C-terminal domain is the NSE/DTE motif. These two motifs flank the entrance of the active site and function in binding a trinuclear magnesium cluster [20,21]. Most terpene synthases belong to monoterpene synthase (MTPSs) [22], sesquiterpene synthase (STPSs), and diterpene synthase (DTPSs) [23]. They all share three conserved domains in the active site, including ‘DDXXD’, ‘DXDD’, and ‘EDXXD’. The ‘R(R)X_8_W’ motif is also essential for monoterpene cyclization, while some MTPSs do not have it [16]. These circumstances can be seen in linalool synthase in rice (*Oryza sativa* L. cv. Nipponbare and Hinohikari) [24]; nerol synthase in soybean (*Glycine max* cv. ‘Bagao’), which has a signal peptide and is believed to be functional in plastid [25]; and FaNES1, the cytosolic terpene synthase identified in strawberry, which is able to use cytosolic GDP and FDP to produce linalool and nerolidiol [26].

*TPS*s in the same subfamilies are similar in sequence and have similar functions. Based on the protein sequence, angiosperm STPSs and DTPSs belong to *TPS-a* subfamily and monoterpene synthases belong to *TPS-b* subfamily. Subfamilies in *TPS-c* and *e/f* have enzyme activities of DTPSs; Gymnosperm-specific *TPS-d* subfamily owns the enzyme activities for MTPSs, STPSs, and DTPSs. *TPS-g* encodes MTPSs, STPSs, and DTPSs that produce mainly acyclic terpenoids. *TPS-h* is *Selaginella moellendorffii*-specific subfamily and putative encodes DTPSs [16,27]. Recently, large amounts of TPSs have been identified by using BLAST and thus used for functional characterization assay to further confirm the activity of TPSs. The functions of TPSs can be mono- or multi-functional, and the enzymes can be highly identical to each other. For instance, the DTPs of levopimaradiene/abietadiene synthase and isopimaradiene synthase showed 91% identity in Norway spruce [28]. Moreover, the functional bifurcation of these two enzymes were proved to be caused by only four amino acid residues [28]. Some TPSs are responsible for producing compounds that are related to plant growth and development, such as gibberellin biosynthesis [29], others are responsible in secondary metabolism like monoterpenes and sesquiterpenes for pollination and defense [30,31]. Molecules catalyzed by *TPS* are usually further modified by cytochromes p450 (CYPs) to generate diverse structures [32].

Orchids show extraordinary morphological, structural, and physiological characteristics unique in the plant kingdom [33]. Containing more than 800 genera and 28,000 species, the Orchidaceae, classified in class Liliopsida, order Asparagales, is one of the largest and most diverse families of flowering plants [33]. They are widely distributed wherever sun shines except Antarctica, and with a variety of life forms from terrestrial to epiphytic [34]. According to molecular phylogenetic studies, Orchidaceae comprises five subfamilies, including Apostasioideae, Cypripedioideae, Vanilloideae, Orchidaideae, and Epidendroideae [35]. Orchids emit various volatile organic compounds (VOCs) to attract their pollinators, and/or the enemy of herbivores for olfactory capture. The emitted VOCs are plant secondary metabolites, and the major natural products include terpenoids, phenylpropenoids, benzeniods, and fatty acid derivatives. The floral scent composed of the VOCs plays an important role in plants, such as pollinator attraction, defense, and plant-to-plant communication, especially in insect-pollinated plants [30,36].

Floral VOCs are characterized into several orchids, including α- and β-pinene for *Cycnoches densiflorum* and *C. dianae* [37]; phenylpropanoids in *Bulbophyllum vinaceum* [38]; α-pinene and *e*-carvone oxide for *Catasetum* integerrimum [39]; *p*-dimethoxybenzene for *Cycnoches ventricosum* and *Mormodes lineata* [39]; β-bisabolene and 1,8-cineole for *Notylia barkeri* [39]; *e*-ocimene and linalool for *Gongora galeata* [39]; monoterpenes in *Orchis mascula* and *Orchis pauciflora* [40]; (Z)-11-eicosen-1-ol in *Dendrobium sinense* [41]; terpenoid of (E)-4,8-dimethylnona-1,3,7-triene (DMNT) in *Calanthe sylvatica* [42] and *Cyclopogon elatus* [43]; (*E*)-β-ocimene and (*E*)-epoxyocimene for *Catasetum cernuum* and *Gongora bufonia* [44]; and farnesol, methyl epi-jasmonate, nerolidol, and farnesene in *Cymbidium goeringii* [45].

*Phalaenopsis* spp. is very popular worldwide for its spectacular flower morphology and colors. Most *Phalaenopsis* orchids are scentless but some do emit scent VOCs [46]. The scented species have been extensively used as breeding parents for the production of scented cultivars, such as *P*. *amboinensis*, *P*. *bellina*, *P*. *javanica*, *P*. *lueddemanniana*, *P*. *schilleriana*, *P*. *stuartiana*, *P*. *venosa*, and *P*. *violace* [47]. *P*. *bellina* and *P*. *violacea* are two scented orchids that are very popular in breeding scented cultivars. *P*. *bellina* emits mainly monoterpenoids, including citronellol, geraniol, linalool, myrcene, nerol, and ocimene [47,48], while *P*. *violacea* emits monoterpenoids accompanied with a phenylpropanoid, cinnamyl alcohol [46]. The VOCs of *P*. *schilleriana* contain monoterpenoids as well, including citronellol, nerol, and neryl acetate [49]. Because of the importance of terpenoids in plants, an increasing number of plants are required to investigate the function and evolution of *TPS*s.

In the present review, we summarized the recent progress in the understanding of the biosynthesis and biological function of terpenoids, and the latest advances in research on the evolution and functional diversification of *TPS*s in Orchidaceae. TPSs from different orchid species are reported to explore the evolutionary history and the evolution diversification of Orchidaceae *TPS*s.

## 2. Terpenoids and Their Biosynthesis in Plants

There are two compartmentalized terpenoid biosynthesis pathways, the mevalonic acid (MVA) pathway that occurs in the cytosol, and the methylerythritol phosphate (MEP) pathway that occurs in plastids to produce isopentenyl diphosphate (IPP) and its allylic isomer-dimethylallyl diphosphate (DMAPP) converted by isopentenyl diphosphate isomerase (IDI) (Figure 1) [50,51,52]. There are four major steps involved in the biosynthesis of terpenoid, beginning with isoprene unit (IPP) formation, which has five carbons. Second, IPP combines to DMAPP by geranyl diphosphate synthase (GDPS), geranylgeranyl diphosphate synthases (GGDPS) or farnesyl diphosphate (FDPS), and generates geranyl diphosphate (GDP), farnesyl diphosphate (FDP) or geranylgeranyl diphosphate (GGDP), respectively [1,27,53,54]. Third, the C_10_-C_20_ diphosphates go through cyclization and rearrangement to produce the basic carbon skeletons for terpenoids catalyzed by *TPS* [53]. The *TPS* family consists of enzymes that use GDP to form cyclic and acylic monoterpenes (C_10_), FDP for sesquiterpene (C_15_), and GGDP for diterpene (C_20_) [16]. Moreover, FDP and GGDP can be dimerized to form the precursors of C_30_ and C_40_. The final step converts terpenes into different skeletons by oxidation, reduction, isomerization, conjugation, and other transformation [53]. *TPS*s are the key enzymes in terpenoid biosynthesis.

## 3. The Evolution of *TPS* Genes in Orchidaceae Species

We chose the whole genome sequences of four orchids, including *A*. *shenzhenica* [54] in Apostasioideae subfamily; *Vanilla planifolia* [55] in Vanilloideae subfamily; and *D*. *catenatum* [56] and *P*. *equestris* [57] in Epidendroideae subfamily. There were two justifications for this selection. First, these four orchids are distributed into three different subfamilies, and their whole genome sequences are available in NCBI (https://www.ncbi.nlm.nih.gov/ (accessed on 6 January 2021).) and OrchidBase database [58] (http://orchidbase.itps.ncku.edu.tw/est/home2012.aspx (accessed on 9 August 2020).). Second, *A*. *ashenzhenica* is the most original orchid, and *P*. *equestris* is the first whole genome sequenced orchid. *V*. *planifolia* produces vanillin and is important in the food industry, and *D*. *catenatum* is a medicinal orchid and produces important secondary metabolites for pharmaceutical purpose. We isolated the *TPS* genes of Orchidaceae through KAAS (http://www.genome.jp/tools/kaas/ (accessed on 21 February 2017).) annotation and BLASTp from the whole genome sequences of four orchids. Each full-length *TPS* is characterized by two conserved domains with Pfam [59] ID PF01397 (N-terminal) and PF03936 (C-terminal) [17]. A total of 9, 27, 35, and 15 *TPS* genes were identified from the whole genome sequences of *A. shenzhenica*, *V. planifolia*, *D. catenatum,* and *P. equestris*, respectively. In addition, *P. aphrodite* with white, scentless flowers and *P. bellina* scented flowers are native species. Their floral transcriptomes are available in Orchidstra and OrchidBase transcriptome database, respectively. 17 *TPS* genes in *P. aphrodite* and 11 *TPS* genes in *P. bellina* were identified from the transcriptome database. The *TPS* genes were denoted with numbers *Ash-*, *KAG-*, *Dca-*, *Peq-*, *PATC-,* and *PbTPS-* identified from *A. shenzhenica*, *V. planifolia*, *D. catenatum*, *P. equestris*, *P. aphrodite,* and *P. bellina*, respectively.

*TPS*s in *P. equestris* and *D. officinale* have been reported [15,60]. These *TPS*s are divided into four subfamilies (*TPS-a*, *TPS-b*, *TPS-c*, and *TPS-e/f*). So, we further investigated *TPS* evolution in Orchidaceae and provided insight into *TPS*s at the genome level. In this review, the encoded amino acid sequences of identified orchid *TPS* genes were aligned with those from *Arabidopsis* and *Abies grandis*, and those from *Selaginella moellendorffii* were used as outgroups (Appendix A Table A1). The phylogenetic tree was constructed using Neighbor-Joining method with Jones–Taylor–Thornton model and pairwise deletion with 1000 bootstrap replicates by using MEGA7 software. The orchid *TPS*s are grouped into *TPS-a*, *-b*, *-e/f*, and *g* subfamilies (Figure 2). Most of the orchid *TPS*s belong to *TPS-a* and *TPS-b* subfamilies (89/115, Table 1). In the *TPS-a* subfamily, copies from dicot and monocot species formed distinct subgroups, which is in accordance to previous studies [15,16]. However, compared to angiosperm dicot species, which have more *TPS*s in *TPS-a* subfamily, orchid (monocot) *TPS*s have more members in *TPS-b* subfamily than in *TPS-a* subfamily. Within *TPS-b* subfamily, these orchid *TPS*s form distinct clades separated from those of *Arabidopsis* (dicot) *TPS*s (Figure 2). Taken together, the persistence of dicot and monocot distinct clades within *TPS-a* and *TPS-b* implies that these *TPS*s have diverged since the ancestor of angiosperm. On the other hand, most of the duplicated orchid *TPS-a* and *TPS-b* copies were species-dependent (i.e., paralogs duplicated within each species). In particular, the number of duplicated orchid *TPS-a* and *TPS-b* copies increased in *V. planifolia* and *D. catenatum* (Figure 2). These data suggest that *TPS-a* and *TPS-b* copies evolved in a species-dependent manner and may have been positively selected to generate exceptionally more multiple copies. *TPS-a* and *TPS-b* are angiosperm-specific subfamilies that are responsible for sesquiterpene or diterpene and monoterpene synthases. These orchid volatile terpenes have critical roles in producing floral scents in order to be attractive to pollinators and to respond to environmental stresses [15]. It is therefore not surprising that *TPS-a* and *TPS-b* subfamilies have diverged greatly in orchid species.

Our phylogenetic analysis also reveals that the orchid *TPS-e/f* subfamily has increased copy numbers compared to that from *A. thaliana* (Table 1; Figure 2). Orchid *TPS-g* subfamily can only be found in *A. shenzhenica* and *V. planifolia* (Table 1; Figure 2), whereas those Epidendroideae *TPS-g* members have perhaps been lost during evolution. There are no orchid *TPS*s in *TPS-c* group that host copalyl diphosphate synthases (CPS) of angiosperm [61]. *TPS-d* and *TPS-h* are gymnosperm and *Selaginella moellendorffii* specific, respectively [16]. Our analysis showed that no orchid *TPS*s were grouped in these subfamilies, in accordance with previous conclusions by Chen et.al, and Trapp et.al. [16,62].

Motifs of identified orchid TPS proteins were predicted using MEME software (https://meme-suite.org/meme/tools/meme (accessed on 19 March 2021).) (Figure 3A), and five major functional conserved motifs of *TPS*s (R(R)X_8_W, EDXXD, RXR, DDXXD, and NSE/DTE) were elucidated (Figure 3B). The *TPS-a* subfamily that encodes STPSs is mainly found in both dicot and monocot plants [9,11,16,63]. In this subfamily, STPSs contain the non-conserved secondary “R” (arginine) of motif R(R)X_8_W that functions in the initiation of the isomerization cyclization reaction [64], or in stabilizing the protein through electrostatic interactions [65]. Compared with *Arabidopsis*, most orchid *TPS*s contain motif R(R)X_8_W, except PATC144727, Peq011664, Dca017107, and PATC155674 in *TPS-a* subfamily (Figure 4A). In contrast, the angiosperm-specific *TPS-b s*ubfamily that encodes MTPSs contains the highly conserved R(R)X_8_W motif. All *TPS*s in *Arabidopsis* TPS-b subfamily contain conserved R(R)X_8_W motif, except AtTPS02 (Figure 4B). However, several members of orchid TPS-b subfamily have lost the conserved R(R)X_8_W motif (Figure 4B). Motifs EDXXD, RXR, DDXXD, and NSE/DTE are highly conserved in *TPS-a* and *-b* subfamilies, while the conserved R(R)X_8_W motif of orchid *TPS*s is divergent in *TPS-b* subfamily.

DTPSs are evolved from kaurene synthase (KS) and CPS. MTPSs and STPSs are evolved from ancestral DTPS through duplication and then sub- or neo-functionalization during evolution [66]. *A. shenzhenica* has clear evidence of whole-genome duplication that is shared by all orchids [54]. Yet, the copies of *TPS* in *A. shenzhenica* are among the fewest and are worthwhile for further investigation. For *Phalaenopsis* orchids, paralogs of *TPS* genes could be identified from each species, implying the duplications were attributed to their common ancestor, and some persisted or lost in current species (Figure 4). For example, *TPS-a* copies of *P. aphrodite*, *P. bellina,* and *P. equestris* species can be found (some lost) in three parallel clades of the phylogenetic tree (*PATC144727/Peq010211/PbTPS02*, *PATC137979/Peq021360,* and *PATC175129/Peq011667*) (red tangle, Figure 4A). Similarly, *TPS-b* copies of *P. aphrodite*, *P. bellina,* and *P. equestris* can be repeatedly identified (some lost) in eight parallel clades, indicating the *TPS-b* gene copy duplications could be traced back to the common ancestor of *Phalaenopsis* species (*PATC208458/Peq006283, PATC153230/PbTPS09, PATC150554/Peq006282, Peq006285/PbTPS07, Peq006275/PbTPS10, PATC127710/Peq013713, PATC068781/Peq013045* and *PATC187424/Peq013048*) (red tangle, Figure 4B).

Members of *TPS-e/f* subfamilies are mainly detected in angiosperm and conifers DTPSs of primary metabolism (i.e., gibberellin biosynthesis) [16,67]. Orchid *TPS-e/f* subfamilies comprise orthologous genes without R(R)X_8_W (Figure 4C), which are consistent with *Arabidopsis*. The *Ash009730* in *TPS-e/f* subfamily, predicted to be KS, was grouped with *KAG0503701* and *Dca000690* (red retangle with red star, Figure 4C). No *TPS*s were found in *A. shenzhenica* in *TPS-f* subclade. As copies of these orchid *TPS-e/f* subfamilies were duplicated within each species, the duplications seem to be species dependent.

*TPS-g* subfamily is closely related to the *TPS-b* but lacks the N-terminal “R(R)X_8_W” motif and encodes MTPSs, STPSs, and DTPSs that produce mainly acyclic terpenoids [68,69]. A highly conserved arginine-rich RXR motif of sesquiterpene synthase reported that the motif is involved in producing a complex with the diphosphate group after the ionization of FPP in sesquiterpene biosynthesis [70]. *TPS-g* subfamily in *Arabidopsis* (AtTPS14) lacks both “R(R)X_8_W” and “RXR” motifs. However, although *TPS*s of *V. planifolia* in *TPS-g* subfamily (those started with KAG in Figure 4D) lack the N-terminal “R(R)X_8_W” motif, they still have the “RXR” motif (Figure 4D). This suggests that *TPS-g* subfamily of *V. planifolia* may have conserved enzyme activities that are capable of accepting a multi-substrate in terpene biosynthesis.

The pharmaceutical effective compounds in *D. catenatum*, a widely used Chinese herb, belong to terpenoid indole alkaloid (TIA) class [71], and many of them contain a terpene group. A sesquiterpene alkaloid-Dendrobine found in *Dendrobium* is believed to be responsible for its medical property [71]. Concomitantly, a significant increased number of *TPS-a TPS*s was detected in *D. catenatum*as as compared to that of other orchid species, which is responsible for sesquiterpene biosynthesis (Table 1). The increased number of *TPS-b* in *Dendrobium* may cause the floral fragrance in *D. catenatum* as well as the formation of TIA. *P. bellina* is a scented orchid with the main floral compounds of monoterpenes including linalool, geraniol, and their derivatives, which attract pollinators [48]. *PbTPS*s from the floral transcriptome database are majorly classified into the *TPS-b* subfamily (Table 1). Previously, the expression of both *PbTPS5* and *PbTPS10* were concomitant with the VOCs (monoterpene linalool and geraniol) emission in *P. bellina* [72]. This suggests that these genes may be involved in the biosynthesis of monoterpene in *P. bellina*. *TPS-e/f* enzymes have diverse functions, including linalool synthase, geranyllinalool synthase, and farnesene synthase in kiwifruit [73,74]. *TPS*s in the *TPS-e/f* subfamily are thought to be dicot-specific because so far no *TPS-e/f* activity has been reported in monocots. However, the number of TPS in *TPS-e/f* expands from 1 in *Apostasia* to 4 in *Phalaenopsis* (Table 1), suggesting that the duplication events of *TPS- b* and *TPS-e/f* have evolved in response to natural selection.

Together, our analyses suggest that orchid **TPS*s* in each subfamily evolved from the early divergence orchid species, such as *A. shenzhenica* and/or *V. planifolia*. The large expansion of *TPS* copies in orchid groups such as *V. planifolia*, *D. catenatum*, and *Phalaenopsis* species might be due to high flexibility for adaptation and evolution through natural selection.

## 4. The Arrangement of *TPS*

The functional cluster phenomenon of *TPS* genes was detected in orchids. Orchid *TPS* gene clusters diverged with tandem or segmental duplications (Figure 5). Tandem duplication inferred that the duplication occurred in the same scaffold, such as *Ash012495* grouped with *Dca000691*/*Dca000692*/*Dca000697* cluster genes in *TPS-b* subfamily (Figure 4B and Figure 5C). *TPS* genes duplicated on different scaffolds is thought to be segmental duplication, *e.x*.: *Ash008718/Ash008719* grouped with two cluster genes of *V. planifolia* (*KAG0458420/KAG0458425/KAG0458429 and KAG0460140/KAG0460156/KAG0460160*) in different scaffolds in the *TPS-g* subfamily (Figure 4D and Figure 5A,B). We identified that 6, 24, 20, and 8 *TPS*s in *A. shenzhenica*, *V. planifolia*, *D*. *catenatum,* and *P. equestris*, respectively, form clusters in the same genome scaffold (Table 2, Figure 5A–D). In addition, these clusters were present with *TPS*s of the same subfamily and therefore the enhancement of functions was predicted. In *A. shenzhenica, V. planifolia*, *D*. *catenatum*, and *P. equestris*, *TPS* genes have three, nine, eight, and three clusters, respectively (Table 2, Figure 5). Each cluster contains two *TPS* genes in *A. shenzhenica*, while more genes are present in the clusters of *V. planifolia*, *D*. *catenatum,* and *P. equestris* (Figure 4). *TPS* genes in the same cluster usually belong to the same subfamily except that *V. planifolia* has one large scaffold containing *TPS* genes of *TPS-a*, *TPS-b,* and *TPS-e/f* subfamilies, yet with huge distance between each subfamily cluster (44 Mb and 5 Mb, respectively). The percentages of clustered *TPS* genes were 66.7%, 81.5%, 57.1%, and 53.3% for *A. shenzhenica*, *V. planifolia, D. catenatum,* and *P. equestris*, respectively, while that was 40.6% in *Arabidopsis thaliana* (Table 2). The cluster density of orchid **TPS*s* could infer the event of *TPS* gene duplication occurred during evolution. The genome sizes of *A. shenzhenica*, *V. planifolia*, *D. catenatum,* and *P. equestris* are 349 Mb, 7449 Mb, 1104 Mb, and 1064 Mb, respectively (Table 3). The cluster densities of **TPS*s* in orchids were 47.3%, 78.6%, 50.5%, and 38.9% for *A. shenzhenica*, *V. planifolia*, *D. catenatum,* and *P. equestris*, respectively (Table 3). Interestingly, orchids have more clusters and higher *TPS* gene density as compared to that of *Arabidopsis*, with that of *V. planifolia* having the highest cluster gene density of *TPS* among the four orchids analyzed. Even though *TPS*s copies of derived orchids (*D. catenatum* and *Phalaenopsis* spp.) were increased compared with those in *A. shenzhenica*, the total number was not linked to the increased genome size.

In plants, gene clusters were often observed for metabolic pathways, such as gene clusters found in oat and *Arabidopsis* related to triterpene biosynthesis pathway [75]. Local duplication of *TPS* gene families in plants has been described and often results in tandem repeats, as an important driver for the expansion [16,76]. The genes related in terpene synthesis are usually lined together, forming functional clusters in plants [77]. The functional clusters of *TPS* genes have already been reported in several plant species, such as *Arabidopsis thaliana* [6], *Vitis vinifera* [9], *Solanum lycopersicum* [77], *Eucalpyus grandis* [78], and rice [79,80]. Genomic clusters of *TPS* genes in *E. grandis* are up to 20 genes [78]. In several *Solanum* species, the gene duplications and divergence give rise to *TPS* gene clusters for terpene biosynthesis [77]. A dense cluster of 45 *V. vinifera TPS*s are present on chromosome 18 [9]. *Arabidopsis TPS* genes are reported with the phenomenon of several gene clusters [6]. In addition, a gene cluster with three *TPS* members, including *Os08g07080*, *Os08g07100*, and *Os08g07120,* is observed in Asian rice *Oryza sativa* and also appears in various rice species including *O*. *glaberrima*, *O*. *rufipogon*, *O*. *nivara*, O. *barthii,* and *O*. *punctata*. [80]. Both conserved and species-specific expression patterns of the clustered rice *TPS*s indicate the functions in insect-damaged plants [80]. The expression of these rice *TPS* genes and their catalytic activities for emission patterns of volatile terpenes is induced by insect damage and is largely consistent [80]. Interestingly, the evolution of *TPS*s with other biosynthesis-related genes was also found to form unexpected connection with time passed. For instance, the evolution of *TPS/CYP* pairs is different in monocot and dicot [81]. *TPS/CYP* pairs duplicate with ancestral *TPS/CYP* pairs as templates to be evolved in dicots, but the evolutionary mechanism of monocot shows that the genome rearrangement of *TPS* and *CYP* occurred independently [81]. In *Solanum* spp., *TPS* forms functional clusters with *cis*-prenyl transferase [77]. Both tandem and segmental duplications significantly contribute toward family expansion and expression divergence and play important roles in the survival of these expanded genes. A functional gene cluster is a group of closely-related genes lined together in a genome, and the study of gene clusters is important for the understanding of evolution within species.

Together, the orchid *TPS* genes formed genomic clusters, and the clusters increased in *V. planifolia* and *D. catenatum*. Combining the results from phylogenetic analysis and functional gene clusters, orchid *TPS*s may be expanded by tandem or segmental duplications. Interestingly, the genome duplication events occurred all the way along the evolution from Apostasioideae to Vanilloideae and Epidendroideae; the *TPS* clusters and copy numbers increased in orchid lineages, such as the early divergence *A. shenzhenica*. The large expansion of orchid *TPS* copies in *V. planifolia*, and *D. catenatum* species might have high flexibility in secondary biosynthesis through natural selection.

## 5. Conclusions

The basic evolution of *TPS* is from duplication and loss of *TPS* genes. In Orchidaceae, we discover that the duplication event of *TPS* occurred among all *TPS* subfamilies. *TPs-a*, *TPS-b,* and *TPS-e/f* subfamilies went through gene duplication, while *TPS-g* duplicated from Apostaceae to Vaniloideae, and then lost from Vaniloideae to Epidendroideae. The driving force of *TPS* evolution in each subfamily may be different. For example, in *TPS-a* and *TPS-b*, the necessity of generating volatile compounds for the interaction of orchids with their pollinators, producing chemical defenses and being responsive to environmental stress, may be the major reason for their rapid evolution. On the other hand, the duplications of *TPS-g* and *TPS-e/f* copies were mainly species dependent and the reason remains to be uncovered.

## Figures and Tables

**Figure 1 ijms-22-06947-f001:**
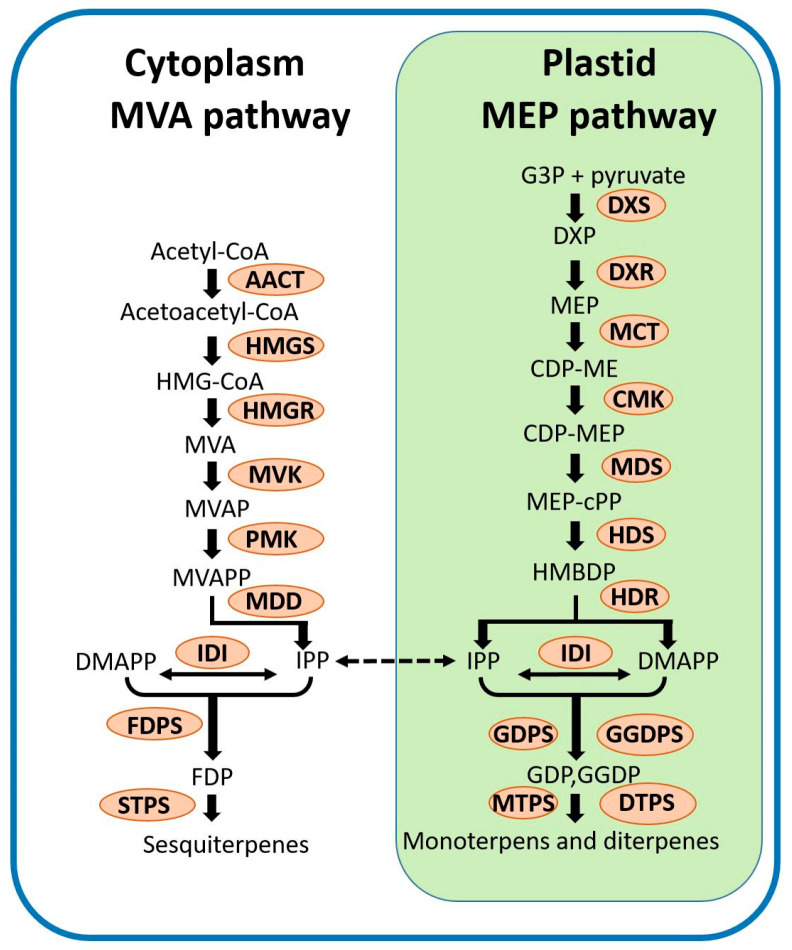
The MVA (left) and MEP (right) pathways responsible for IPP and DMAPP biosynthesis and monoterpene biosynthesis in plants. AACT, acetoacetyl-CoA thiolase; CMK, 4-(cytidine 5′ -diphospho)-2-*C*-methyl-d-erythritol kinase; DMAPP, dimethylallyl diphosphate; DXR, 1-deoxy-d-xylulose 5-phosphate reductoisomerase; DXS, 1-deoxyd- xylulose 5-phosphate synthase; FDP, farnesyl diphosphate; FPPS, farnesyl diphosphate synthase; G3P, d-glyceraldehyde 3-phosphate; GDPS, geranyl diphosphate synthase; GDP, geranyl diphosphate; HDR, (*E*)-4-hydroxy-3-methylbut-2-enyl diphosphate reductase; HDS, (*E*)-4-hydroxy-3-methylbut-2-enyl diphosphate synthase; HMGR, 3-hydroxy-3-methylglutaryl-CoA reductase; HMGS, 3-hydroxy-3-methylglutaryl- CoA synthase; IDI, isopentenyl diphosphate isomerase; IPP, isopentenyl diphosphate; MCT, 2-*C*-methyl-d-erythritol 4-phosphate cytidylyltransferase; MDD, mevalonate diphosphate decarboxylase; MDS, 2-*C*-methyld-erythritol 2,4-cyclodiphosphate synthase; MVK, mevalonate kinase; MVAP, mevalonate 5-phosphate; MVAPP, mevalonate diphosphate; PMK, phosphomevalonate kinase; TPS, terpene synthase.

**Figure 2 ijms-22-06947-f002:**
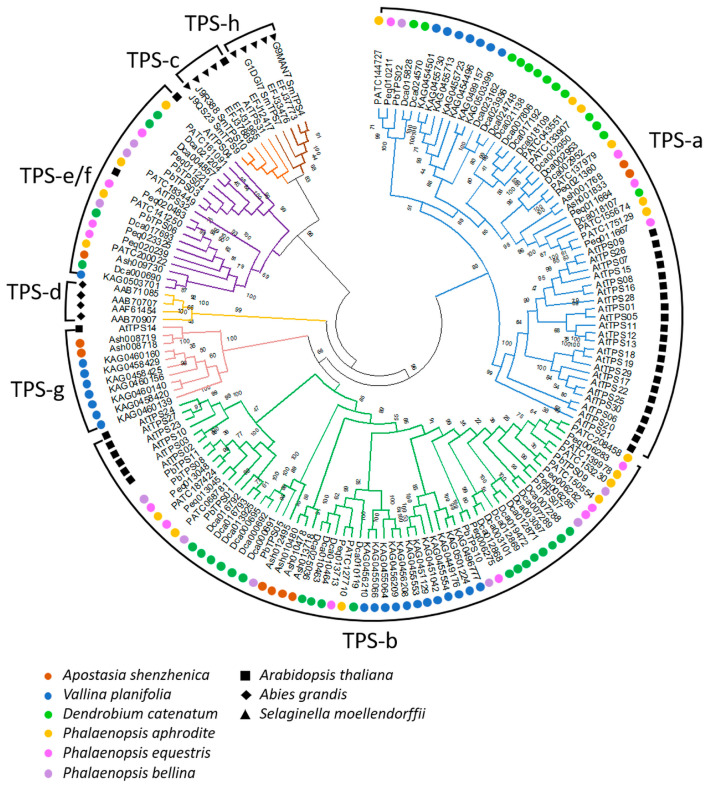
Phylogenetic analysis of terpene synthases. *TPS*s in Orchidaceae, including *A. shenzhenica*; *V. planifolia*; *D. catenatuml*
*P. equestris*
*Phalaenopsis aphrodite*; *P. bellina*, *Arabidopsis thaliana*, and *Abies grandis*; and *S. moellendorffii* were used. Sequence analysis was performed using MEGA 7.0 to create a tree using the nearest neighbor-joining method. The coding sequence was used for analysis. The numbers at each node represent the bootstrap values. Various colors mean distinct subfamilies and special symbols represent different plant species, with solid circles, tangle, diamond, and triangle illustrating Orchidaceae, *Arabidopsis thaliana*, *A. grandis*, and *S. moellendorffii*, respectively.

**Figure 3 ijms-22-06947-f003:**
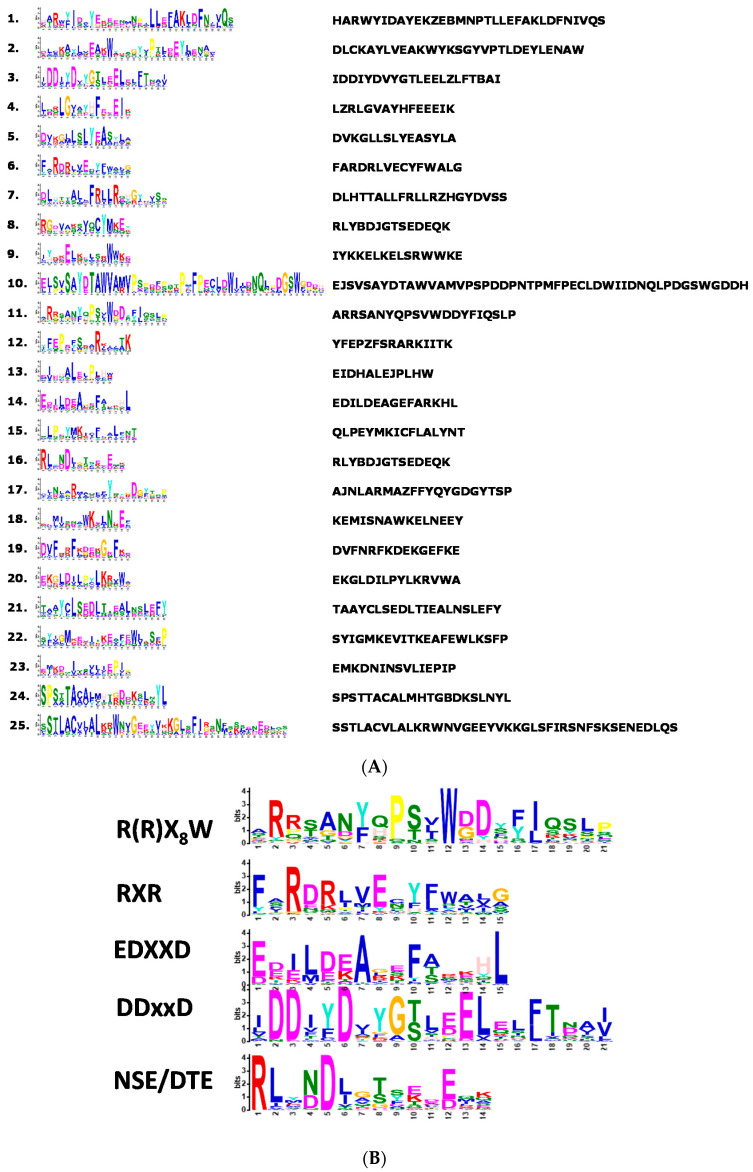
The amino acid sequences of the predicted motifs in *TPS* proteins. (**A**) Twenty-five classical motifs in *TPS* proteins were analyzed using the MEME tool. The width of each motif ranges from 6 to 50 amino acids. The font size represents the strength of conservation. (**B**) The amino acid sequences of five highly conserved motifs in *TPS* proteins.

**Figure 4 ijms-22-06947-f004:**
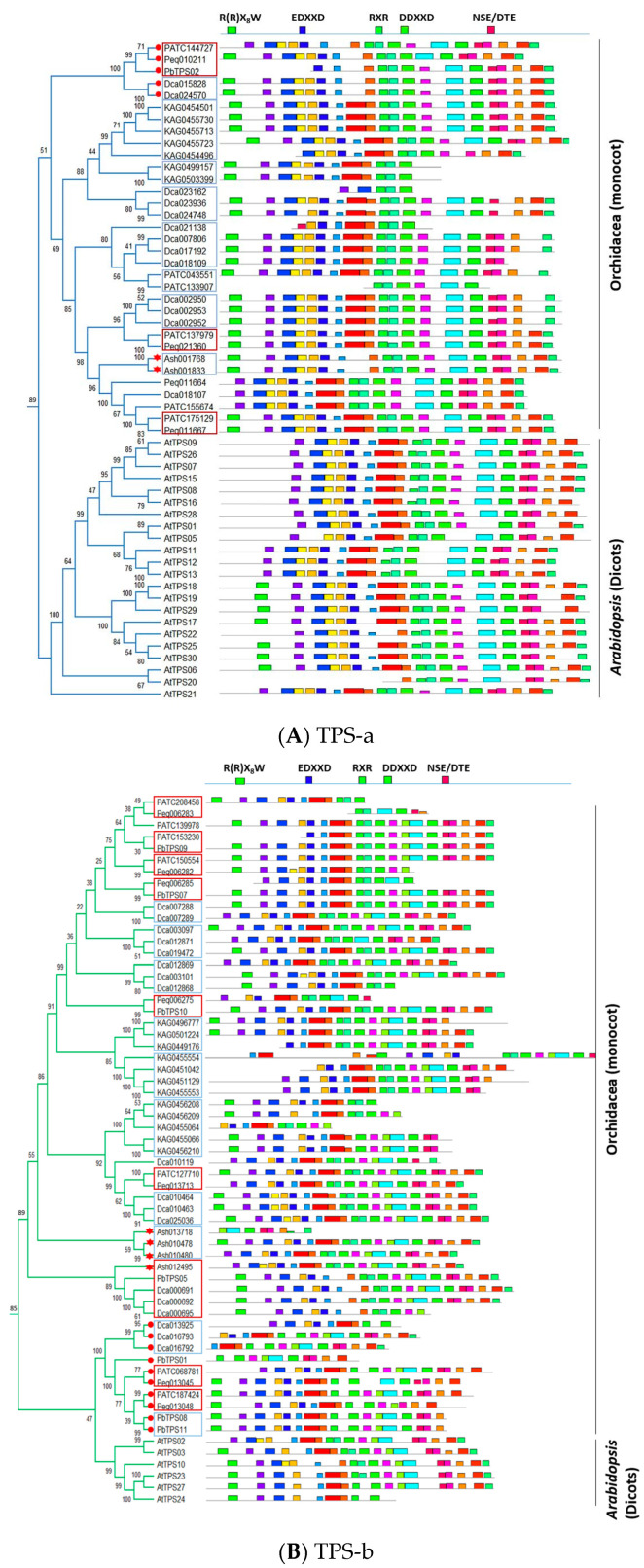
Motif structures of *TPS* proteins. (**A**–**D**) are *TPS-a, -b, -e/f*, and *-g* subfamilies, respectively. Twenty-five classical motifs in *TPS* proteins were analyzed by using the MEME tool. The width of each motif ranged from 6 to 50 amino acids. Different color blocks represent distinct motifs. Star indicates *TPS*s of *A. shenzhenica*, and the red solid circle indicates the out group of *Apostasia TPS*s. The red and blue rectangle squares reveal orthologous and paralogous gene pairs, respectively.

**Figure 5 ijms-22-06947-f005:**
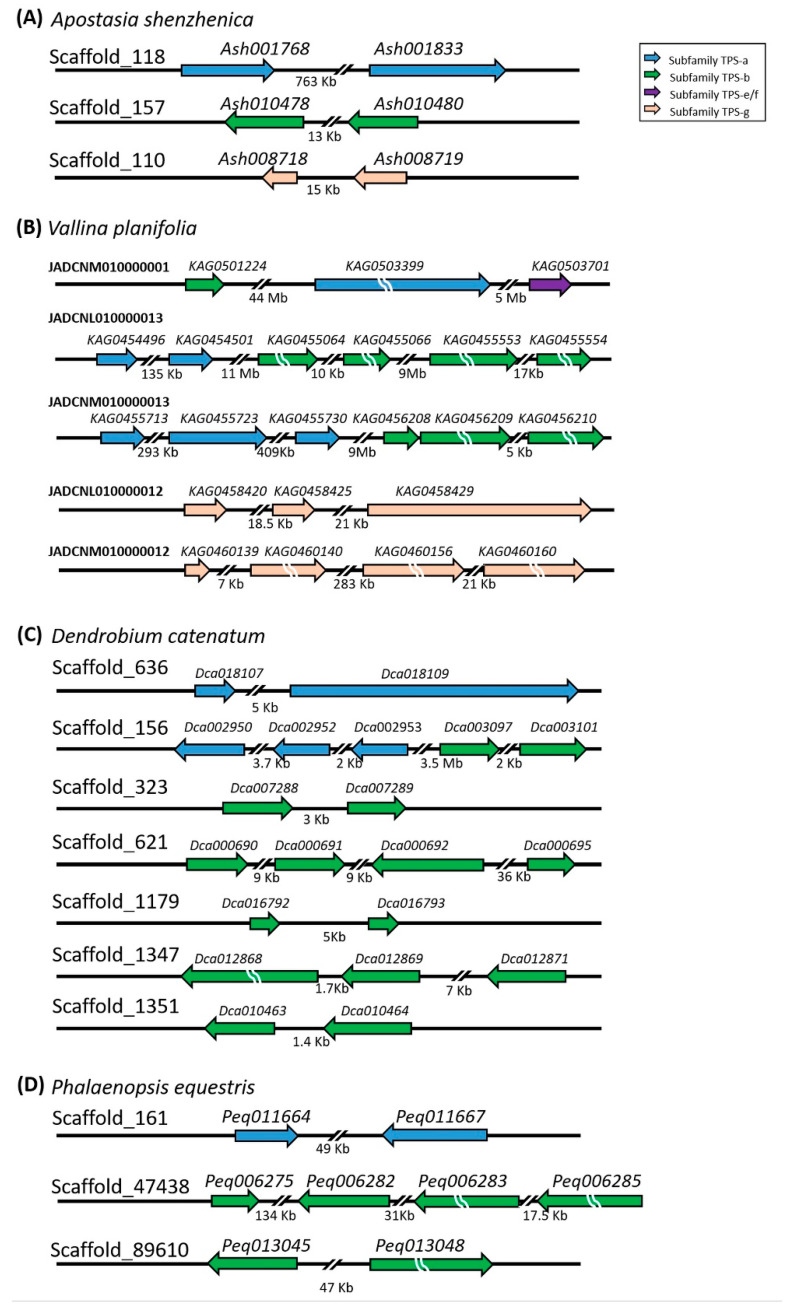
Gene clusters in Orchidaceae genome. Clustered genes in the genomic scaffolds of *A. shenzhenica* (**A**), *V. planifolia* (**B**), *D. catenatum* (**C**), and *P. equestris* (**D**), respectively. The *TPS* genes located on the scaffolds are identified from the assembled whole genome sequences of *A. shenzhenica*, *V. planifolia*, *D. catenatum,* and *P. equestris*. The direction of arrows illustrates the forward translation of genes in the scaffolds. Various colors indicate the distinct *TPS* subfamilies. Blue, green, purple, and bisque colors represent *TPS* genes in *TPS-a*, *-b*, *-e/f*, and *-g* subfamilies, respectively. Break lines indicate the shrink length of genes.

**Table 1 ijms-22-06947-t001:** The number of **TPS*s* subfamilies in Orchidaceae and other plant species.

	*TPS* Subfamily								
Species	*a*	*b*	*c*	*d*	*e/f*	*g*	*h*	Total	Reference
*Apostasia shenzhenica*	2	4	0	0	1	2	0	9	This research
*Vallina planifolia*	7	12	0	0	1	7	0	27	This research
*Dendrobium catenatum*	13	18	0	0	4	0	0	35	This research
*Phalaenopsis equestris*	4	7	0	0	4	0	0	15	This research
*Phalaenopsis aphrodite*	6	7	0	0	4	0	0	17	This research
*Phalaenopsis bellina*	1	7	0	0	3	0	0	11	This research
*Arabidopsis thaliana*	22	6	1	0	2	1	0	32	Aubourg et al. (2002) [6]
*Solanum lycopersicum*	12	8	2	0	5	2	0	29	Falara et al. (2011) [10]
*Oryza sativa*	18	0	3	0	9	2	0	32	Chen et al. (2014) [14]
*Sorghum bicolor*	15	2	1	0	3	3	0	24	Paterson et al. (2009) [8]
*Vitis vinifera*	30	19	2	0	1	17	0	69	Martin et al. (2010) [9]
*Populus trichocarpa*	16	14	2	0	3	3	0	38	Irmisch et al., (2014) [13]
*Selaginella moellendorffii*	0	0	3	0	3	0	8	14	Li et al., (2012) [11]

**Table 2 ijms-22-06947-t002:** The gene clusters of **TPS*s* in the genome of Orchidaceae and *Arabidopsis thaliana*.

Species	Number of Clusters	Number of Scaffolds	Number of Clustered *TPS*s	Number of Total *TPS*s	Percentage of Clustered *TPS*s (%)
*Apostasia shenzhenica*	3	3	6	9	66.7
*Vallina planifolia*	7	5	22	27	81.5
*Dendrobium catenatum*	8	7	20	35	57.1
*Phalaenopsis equestris*	3	3	8	15	53.3
*Arabidopsis thaliana* [6]	5	5	13	32	40.6

**Table 3 ijms-22-06947-t003:** The gene density of **TPS*s* in the genome of Orchidaceae and other plant species.

Species	Genome Size (Mb)	Cluster Length of *TPS*s (Kb)	Total Length of *TPS*s (Kb)	Cluster Density of *TPS*s (%)
*Apostasia shenzhenica*	349	26	56	47.3
*Vallina planifolia*	744	595	758	78.6
*Dendrobium catenatum*	1104	125	248	50.5
*Phalaenopsis equestris*	1064	62	158	38.9
*Arabidopsis thaliana*	120	43	109	39.9

## Data Availability

Not applicable.

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
