# Peer review of "Evolution of Terpene Synthases in Orchidaceae"

_ijms, 2021, doi:10.3390/ijms22136947_

Round 1

Reviewer 1 Report

The manuscript titled “Evolution of terpene synthases in Orchidaceae”, by Huang et al., submitted for publication in the International Journal of Molecular Sciences is interesting to the Journal and scientists. In the study, the authors used bioinformatics software and made a comprehensive phylogenetic analysis of terpene synthases (TPSs) in various Orchidaceae species.

Ms needs a minor revision.

The abstract should be rewritten. You should first write general info about TPSs, then why are they important, and why Orchidaceae species were selected for analysis. There is no answer to the question of why this issue is scientifically important. Some kind of support is given in the first paragraph of the Introduction. In my opinion, it should already be there.

The Introduction section is written very well. It contains a detailed introduction to the content of the article.

“Identification of TPS genes in Orchidaceae species

169 In this review, we have isolated the TPS genes of Orchidaceae through KAAS 170 (http://www.genome.jp/tools/kaas/) annotation and BLASTp from the whole genome sequences of four orchids.”

Please explain why these four species were chosen for analysis? Maybe they are significant crops, produce important metabolites, etc.

Figures 1, 2, 3, 4, and 5 require a significant quality adjustment.

Author Response

Point1: The abstract should be rewritten. You should first write general info about TPSs, then why are they important, and why Orchidaceae species were selected for analysis. There is no answer to the question of why this issue is scientifically important. Some kind of support is given in the first paragraph of the Introduction. In my opinion, it should already be there.

Response 1: Thank you for the suggestion. The abstract is amended as follows: “Terpenoids are the largest class of plant secondary metabolites and are one of the major emitted volatile compounds released to the atmosphere. They play functions of attracting pollinators or defense function, insecticidal properties and even for pharmaceutical agents. Because of the importance of terpenoids, an increasing number of plants are required to investigate the function and evolution of terpene synthases (TPSs) that are the key enzymes in terpenoids biosynthesis. Orchidacea containing more than 800 genera and 28,000 species, is one of the largest and most diverse families of flowering plants, and widely distributed. Here, the diversification of the TPSs evolution in Orchidaceae is revealed. A characterization and phylogeny of TPSs from 4 different species with whole genome sequence available. Phylogenetic analysis of orchid TPSs indicates these genes divided into TPS-a, -b, -e/f and g subfamilies and their duplicated copies are increased in derived orchid species compared to that in the early divergence orchid, A. shenzhenica. The large increase of both TPS-a and TPS-b copies can probably be attributed to the pro-duction of different volatile compounds for attracting pollinators or generating chemical defenses in derived orchid lineages; while the duplications of TPS-g and TPS-e/f copies were oc-curred in species dependent manner.” (Please see the Abstract in the amended revision.)

Point 2: “Identification of TPS genes in Orchidaceae species”.

169 In this review, we have isolated the TPS genes of Orchidaceae through KAAS 170 (http://www.genome.jp/tools/kaas/) annotation and BLASTp from the whole genome sequences of four orchids.”

Please explain why these four species were chosen for analysis? Maybe they are significant crops, produce important metabolites, etc.

Response 2: We chose the whole genome sequences of four orchids, including A. shenzhenica in Apostasioideae subfamily, Vanilla planifolia in Vanilloideae subfamily, D. catenatum, and P. equestris in Epidendroideae subfamily. There were two justifications for this selection. First, these four orchids are distributed in 3 different subfamilies, and their whole genome sequences are available in NCBI (https://www.ncbi.nlm.nih.gov/) and OrchidBase (http://orchidbase.itps.ncku.edu.tw /est/home2012.aspx). Second, A. ashenzhenica is the most original orchid, and P. equestris is the first orchid with whole genome sequenced. V. planifolia produces vanillin and is important in food industry, and D. catenatum is a medicinal orchid and produces important secondary metabolites for pharmaceutical purpose.  (Please lines 159-167 in the amended revision.)

Point 3: Figures 1, 2, 3, 4, and 5 require a significant quality adjustment.

Response 3: Thank you for the suggestion. We have improved the quality of all the figures. (Please see figures in the amended revision.)

Reviewer 2 Report

Dear Authors,

The ms presented for review is dedicated to terpenoids in Orchidaceae. Starting from the very beginning the Authors describe their role in plants, biosynthesis, and functions. Till line 183 the ms sounds like a real review and was interesting to me as it gives a good overview of the terpenoids' role in general.

However, everything is changing from that point. In fact, the ms is a kind of Authors' bioinformatic study on the TPS in Orchidaceae species. I should stress that this part of the ms is the most interesting one and should be published as a separate issue.

Maybe I am wrong, but a review paper should not be a kind of original study (or at least it should be limited to the minimum).

Concerning language, I do not think I am the right person to judge about English but please, verify line 98 (produce ore generate?) and 141 (ПTPSs - what is the П?).

My guess is that the ms should be divided into two separate papers. The first one should be a review whereas the other one the original paper dedicated to the TPS genes. I would like to stress that both parts are interesting and worth being published. Alternatively, please try to reduce the first 183 lines and write the part as an introduction. 

Thus, unless it is allowed by the Journal to publish a review paper in such a form, I would recommend considering a major revision of the ms.

Author Response

Point 1: The ms presented for review is dedicated to terpenoids in Orchidaceae. Starting from the very beginning the Authors describe their role in plants, biosynthesis, and functions. Till line 183 the ms sounds like a real review and was interesting to me as it gives a good overview of the terpenoids' role in general. However, everything is changing from that point. In fact, the ms is a kind of Authors' bioinformatic study on the TPS in Orchidaceae species. I should stress that this part of the ms is the most interesting one and should be published as a separate issue. Maybe I am wrong, but a review paper should not be a kind of original study (or at least it should be limited to the minimum).

Response 1: Thank you for your reminding. In fact, this is a review article, since genome-wide investigation of TPSs have been published for several orchids. For example, TPSs in P. equestris and D. officinale have been reported (Tsai et al., 2017; Yu et al., 2020) These TPSs are divided into four subfamilies (TPS-a, TPS-b, TPS-c, and TPS-e/f). So, we further investigated TPS evolution in Orchidaceae and provide insight of TPSs in the genome level. (Please see lines 179-181 in the amended revision.)

References:

  1. Tsai, W.C., Dievart, A., Hsu, C.C., Hsiao, Y.Y., Chiou, S.Y., Huang, H. and Chen, H.H. 2017. Post genomics era for orchid research. Stud. 58, 61.
  2. Yu, Z., Zhao, C., Zhang, G., Teixeira da Silva, J.A. and Duan, J., 2020. Genome-wide identification and expression profile of TPS gene family in Dendrobium officinale and the role of DoTPS10 in linalool biosynthesis. J. Mol. Sci. 21, 5419.

Point 2: Concerning language, I do not think I am the right person to judge about English but please, verify line 98 (produce ore generate?) and 141 (ПTPSs - what is the П?).

Response 2: Thank you for the reminding. In line 98, the sentence is “Second, IPP combine to DMAPP by geranyl diphosphate synthase (GDPS), geranylgeranyl diphosphate synthases (GGDPS) or farnesyl diphosphate (FDPS), and generate geranyl diphosphate (GDP), farnesyl diphosphate (FDP) or geranylgeranyl diphosphate (GGDP), respectively”. In line 141, the sentence “Class-ПTPSs contain a conserved ‘DXDD’ motif for protonation-initiation cyclization of GDP to CDP” is removed from the amended version. (Please see lines 129-132 in the amended revision.)

Point 3: My guess is that the ms should be divided into two separate papers. The first one should be a review whereas the other one the original paper dedicated to the TPS genes. I would like to stress that both parts are interesting and worth being published. Alternatively, please try to reduce the first 183 lines and write the part as an introduction.

Response 3: Thank you. We take the second suggestion by reducing the first 183 lines into part of the Introduction. (Please see lines 52-67 and 74-84 in the amended revision.)

Point 4: Thus, unless it is allowed by the Journal to publish a review paper in such a form, I would recommend considering a major revision of the ms.

Response 4: The manuscript has been reorganized and revised accordingly. (Please see the amended revision.)

Round 2

Reviewer 2 Report

Dear Authors,

Thank you for considering my suggestions making changes in the ms. 

Some minor suggestions:

Line 131: produce generate – which one? produce or generate?

Line 298: …the enzyme activity for capable of 298 multi-substrate in terpene biosynthesis. Is it correct?

Author Response

Point 1: Line 131: produce generate – which one? produce or generate?

Response 1: Thank you for your reminding. We amended the sentence as “ Second, IPP combine to DMAPP by geranyl diphosphate synthase (GDPS), geranylgera-nyl diphosphate synthases (GGDPS) or farnesyl diphosphate (FDPS), and generate geranyl diphosphate (GDP), farnesyl diphosphate (FDP) or geranylgeranyl diphos-phate (GGDP), respectively.” (Please see lines 129-132 in amended version)

Point 2: Line 298: …the enzyme activity for capable of 298 multi-substrate in terpene biosynthesis. Is it correct?

Response 2: Thank you for your reminding. We amended the sentence as “ This suggests that TPS-g subfamily of V. planifolia may have conserved enzyme activities for capable of accepting multi-substrate in terpene biosynthesis.” (Please see lines 299-301 in amended version)